# Occlusion-Free Road Segmentation Leveraging Semantics for Autonomous Vehicles

**DOI:** 10.3390/s19214711

**Published:** 2019-10-30

**Authors:** Kewei Wang, Fuwu Yan, Bin Zou, Luqi Tang, Quan Yuan, Chen Lv

**Affiliations:** 1Hubei Key Laboratory of Advanced Technology for Automotive Components, Wuhan University of Technology, Wuhan 430070, China; wkw199q@whut.edu.cn (K.W.); yanfuwu@vip.sina.com (F.Y.); tlqqidong@163.com (L.T.); 231943@whut.edu.cn (Q.Y.); 2Hubei Collaborative Innovation Center for Automotive Components Technology, Wuhan University of Technology, Wuhan 430070, China; 3Hubei Research Center for New Energy & Intelligent Connected Vehicle, Wuhan 430070, China; 4School of Mechanical and Aerospace Engineering, Nanyang Technological University, 639798, Singapore; lyuchen@ntu.edu.sg

**Keywords:** autonomous vehicles, scene understanding, occlusion reasoning, road detection

## Abstract

The deep convolutional neural network has led the trend of vision-based road detection, however, obtaining a full road area despite the occlusion from monocular vision remains challenging due to the dynamic scenes in autonomous driving. Inferring the occluded road area requires a comprehensive understanding of the geometry and the semantics of the visible scene. To this end, we create a small but effective dataset based on the KITTI dataset named KITTI-OFRS (KITTI-occlusion-free road segmentation) dataset and propose a lightweight and efficient, fully convolutional neural network called OFRSNet (occlusion-free road segmentation network) that learns to predict occluded portions of the road in the semantic domain by looking around foreground objects and visible road layout. In particular, the global context module is used to build up the down-sampling and joint context up-sampling block in our network, which promotes the performance of the network. Moreover, a spatially-weighted cross-entropy loss is designed to significantly increases the accuracy of this task. Extensive experiments on different datasets verify the effectiveness of the proposed approach, and comparisons with current excellent methods show that the proposed method outperforms the baseline models by obtaining a better trade-off between accuracy and runtime, which makes our approach is able to be applied to autonomous vehicles in real-time.

## 1. Introduction

Reliable perception of the surrounding environment plays a crucial role in autonomous driving vehicles, in which robust road detection is one of the key tasks. Many types of road detection methods have been proposed in the literature based on monocular camera, stereo vision, or LiDAR (Light Detector and Ranging) sensors. With the rapid progress in deep learning techniques, significant achievements in segmentation techniques have significantly promoted road detection in monocular images [1,2,3,4,5]. Generally, these algorithms label each and every pixel in the image with one of the object classes by color and textual features. However, the road is often occluded by dynamic traffic participants as well as static transport infrastructures when measured with on-board cameras, which makes it hard to directly obtain a full road area. When performing decision-making in extremely challenging scenarios, such as dynamic urban scenes, a comprehensive understanding of the environment needs to carefully tackle the occlusion problem. As to the road detection task, road segmentation of the visible area is not sufficient for path planning and decision-making. It is necessary to get the whole structure and layout of the local road with an occlusion reasoning process in complex driving scenarios where clutter and occlusion occur with high frequency.

Inspired by the fact that human beings are capable of completing the road structure in their minds by understanding the on-road objects and the visible road area, we believe that a powerful convolution network could learn to infer the occluded road area as human beings do. Intuitively, to the occlusion reasoning task, the color and texture features are of relatively low importance, what matters is the semantic and spatial features of the elements in the environment. As far as we know, semantic segmentation [6,7,8] is one of the most complete forms of visual scene understanding, where the goal is to label each pixel with the corresponding semantic label (e.g., tree, pedestrian, car, etc.). So, instead of an RGB image, we performed the occlusion reasoning road segmentation using semantic representation as input, which could be obtained by popular deep learning methods in real applications or human-annotated ground truth in the training phase. As shown in Figure 1, traditional road segmentation takes RGB image as input and labels road only in the visible area. As a comparison, our proposed occlusion-free road segmentation (OFRS) intends to leverage the semantic representation to infer the occluded road area in the driving scene. Note that the semantic input in the figure is just a visualization of the semantic representation, the actual input is the one-hot type of semantic label.

In this paper, we aim to infer the occluded road area utilizing the semantic features of visible scenes and name this new task as occlusion-free road segmentation. First, a suitable dataset is created based on the popular KITTI dataset, which is referred to as the KITTI-OFRS dataset in the following. Second, an end-to-end lightweight and efficient fully convolutional neural networks for the new task is proposed to learn the ability of occlusion reasoning. Moreover, a spatially-dependent weight is applied to the cross-entropy loss to increase the performance of our network. We evaluate our model on different datasets and compare it with some other excellent algorithms which pursue the trade-off between accuracy and runtime in the semantic segmentation task. 

The main contributions of this paper are as follows:We analyze the occlusion problem in road detection and propose the novel task of occlusion-free road segmentation in the semantic domain, which infers the occluded road area using semantic features of the dynamic scenes. To complete this task, we create a small but efficient dataset based on the popular KITTI dataset named the KITTI-OFRS dataset, design a lightweight and efficient encoder–decoder fully convolution network referred to as OFRSNet and optimize the cross-entropy loss for the task by adding a spatially-dependent weight that could significantly increase the accuracy.We elaborately design the architecture of OFRSNet to obtain a good trade-off between accuracy and runtime. The down-sampling block and joint context up-sampling block in the network are designed to effectively capture the contextual features that are essential for the occlusion reasoning process and increase the generalization ability of the model.

The remainder of this paper is organized as follows: First, the related works in road detection are briefly introduced in Section 2. Section 3 introduces the methodology in detail, and Section 4 shows the experimental results. Finally, we draw conclusions in Section 5.

## 2. Related Works

Road detection in autonomous driving has benefited from the development of the deep convolutional neural networks in recent years. Generally, the road is represented by its boundaries [9,10] or regions [1,2,11]. Moreover, road lane [12,13,14] and drivable area [15,16] detection also attract much attention from researchers, which concern the ego lane and the obstacle-free region of the road, respectively. The learning-based methods usually outperform the model-based methods due to the developed segmentation techniques. The model-based methods identify the road structure and road areas by shape [17,18] or appearance models [19]. The learning-based methods [3,6,7,16,20,21] classify the pixels in images as road and non-road, or road boundaries and non-road boundaries. 

However, the presence of foreground objects makes it hard to obtain full road despite the occlusion. To infer the road boundaries despite the occlusion, Suleymanov et al. [22] presented a convolutional neural network that contained intra-layer convolutions and produced outputs in a hybrid discrete-continuous form. Becattini et al. [23] proposed a GAN-based (Generative Adversarial Network) semantic segmentation inpainting model to remove all dynamic objects from the scene and focus on understanding its static components (such as streets, sidewalks, and buildings) to get a comprehension of the static road scene. In contrast to the above solutions, we conduct occlusion-free road segmentation to infer the occluded road area as a pixel-wise classification task.

Even though the deep-learning methods have achieved remarkable performance in the pixel-wise classification task, to achieve the best trade-off between accuracy and efficiency is still a challenging problem. Vijay et al. [20] presented a novel and practical deep fully convolutional neural network architecture for semantic pixel-wise segmentation termed SegNet, which follows encoder–decoder architecture that is designed to be efficient both in memory and computational time in inference phase. Adam et al. [24] proposed a fast and compact encoder–decoder architecture named ENet that significantly has fewer parameters, and provides similar or better accuracy to SegNet. Romera et al. [25] proposed a novel layer design that leverages skip connections and convolutions with 1D kernels, which highly reduces the compute cost and increase the accuracy. Inspired by these networks, we follow the encoder–decoder architecture and enhance the down-sampling and up-sampling blocks with contextual extraction operations [26,27,28], which are proved to be helpful for segmentation-related tasks. This contextual information is even more essential and effective for our occlusion reasoning task, which needs a comprehensive understanding of the driving scenes.

## 3. Occlusion-Free Road Segmentation

### 3.1. Task Definition

The occlusion-free road segmentation task is defined as a pixel-level classification as the traditional road segmentation but with occlusion reasoning process to obtain a full representation of the road area. The input is fed to the model as a one-hot encoded tensor of the semantic segmentation labels or predicted semantic segmentation probabilities’ tensor $I\in {\left[0,1\right]}^{W\times H\times C}$, where W is the width of the image, H its height, and C the number of classes. In the same way, we trained the network to output a new tensor $O\in {\left[0,1\right]}^{W\times H\times 2}$ with the same width and height but containing only two categories belonging to road and non-road. 

### 3.2. Network Architecture

The proposed model is illustrated in Table 1 and visualized in Figure 2, and was designed to get the best possible trade-off between accuracy and runtime. We followed the current trend of using convolutions with residual connections [29] as the core elements of our architecture, to leverage their success in classification and segmentation problems. Inspired by SegNet and ENet, an encoder–decoder architecture was adopted for the whole network architecture. The residual bottleneck blocks of different types were used as the basic blocks in the encoder and decoder. Dilated convolution was applied in the blocks to enlarge the respective field of the encoder. What is more, the context module was combined with regular convolution to obtain a global understanding of the environment, which is really essential to infer the occluded road area. In the decoder, we proposed a joint context up-sampling block to leverage the features of different resolutions to obtain richer and global information.

**Context Convolution Block** Recent works have shown that contextual information is helpful for models to predict high-quality segmentation results. Modules which could enlarge the receptive field, such as ASPP [21], DenseASPP [30], and CRFasRNN [31], have been proposed in the past years. Most of these works explore context information in the decoder phase and ignore the surrounding context when encoding the features in the early stage. On the other hand, the attention mechanism has been widely used for increasing model capability. Inspired by the non-local block [27] and SE block [26], we proposed the context convolution, as shown in Figure 3. A context branch from [28] was added, bypassing the main branch of the convolution operation. As can be seen in Equation (1), the context branch first adopted a 1 × 1 convolution $W_{k}$ and softmax function to obtain the attention weights, and then performed the attention pooling to obtain the global context features; then the global context features were transformed via a 1 × 1 convolution $W_{\upsilon }$ and was added to the features of the main convolution branch.
(1)$$z_{i}=x_{i}+W_{\upsilon }\sum_{j=1}^{N_{p}}\frac{\exp\left(W_{k}x_{j}\right)}{\sum_{m=1}^{N_{p}}\exp\left(W_{k}x_{m}\right)}x_{j}\;,$$
where $W_{k}$ and $W_{\upsilon }$ denote linear transformation matrices.

**Down-Sampling Block** In our work, the down-sampling block performed down-sampling by using a 3 × 3 convolution with stride 2 in the main branch of a context convolution block, as stated above. The context branch extracted the global context information to obtain a global understanding of features. Down-sampling lets the deeper layers gather more context (to improve classification) and helps to reduce computation. And we used two down-sampling blocks at the start of the network to reduce the feature size and make the network works efficiently for large input. 

**Joint Context Up-Sampling Block** In the decoder, we proposed a joint context up-sampling block, which takes two feature maps from different stages in the encoder, as shown in Figure 4. The feature map from the earlier stage with bigger resolution and fewer channels carry sufficient details in spatial, and the feature map from the later stage with a smaller resolution and more channels contain the necessary facts in context. The joint context up-sampling block combines these two feature maps gently and efficiently using a context convolution block and bilinear up-sampling. The two branches of the two feature maps were concatenated along the channels, and a context convolution block was applied to the concatenated feature map. As shown in Figure 2, the joint context up-sampling blocks follow a sequential architecture, the current block utilized the former results and the corresponding decoder features, which made the up-sampling operation more effective. 

**Residual Bottleneck Blocks** Between the down-sampling and up-sampling blocks, some residual blocks were inserted to perform the encoding and decoding. In the early stage of the encoder, we applied factorized residual blocks to extract dense features. As shown in Figure 5b, a 3 × 3 convolution was replaced by a 3 × 1 convolution and a 1 × 3 convolution in the residual branch to reduce parameters and computation. In the later stage of the encoder, we stacked dilated convolution blocks with different rates to obtain a larger receptive field and obtain more contextual information. The dilated convolution block applied a dilated convolution on the 3 × 3 convolution in the residual branch compared to the regular residual block, as shown in Figure 5c. The dilate rates in the stacked dilated residual blocks were 1, 2, 5, and 9, which were carefully chosen to avoid the gridding problem when inappropriate dilation rate is used. One dilated residual block consisted of two groups of stacked dilated residual blocks in our network. In the decoder phase, two continuous regular residual blocks were inserted between the joint context up-sampling blocks.

### 3.3. Loss Function

As to the classification tasks, the cross-entropy loss has proved very effective. However, in our task, the road edge area needs more attention paid to it when performing the inference process, and the faraway road in the image took fewer pixels. We proposed a spatially-dependent weight to handle this problem to enhance the loss on the road edge region and faraway road area. The road edge region (ER) was defined as a set of the pixels around the road edge pixels E, which was obtained from the ground truth label image using the Canny algorithm [32], as shown in Figure 6. The Manhattan distance was adopted to calculate the distance between other pixels and edge pixels, and $T_{w}\in R$ was used to control the region size. Then the weight is defined as Equation (3), which takes into account the road edge region and the faraway distance factor. The loss function with spatial weight is shown in Equation (4), which is referred to as CE-SW, and the traditional cross-entropy loss is referred to as CE in our paper. The experiment showed that the CE-SW could significantly improve the performance of the models on the occlusion-free road segmentation task.
(2)$$\mathrm{ER}=\{v\left(i^{\prime },j^{\prime }\right)|\;\left|i-i^{\prime }\right|+\left|j-j^{\prime }\right|<T_{w},\;e\left(i,j\right)\in E,\;v\left(i^{\prime },j^{\prime }\right)\in \mathrm{Img}\}\;,$$
(3)$$w\left(i,j\right)=\{\begin{array}{c} 1,\;\;if\;p\left(i,j\right)\in \bar{ER}\\ \frac{k*\left|i-i_{0}\right|+\left|j-j_{0}\right|}{k*h+w/2}*2+2,\;\;if\;p\left(i,j\right)\in ER \end{array}\;,$$
where w and *h* are the width and height of the image, *k=h/w* is the rate to balance the height and width of the image, i and j are the pixel index, $i_{0}$ and $j_{0}$ the bottom center pixel index.
(4)$$\mathrm{Loss}\left(y,p\right)=\sum_{i}^{H}\sum_{j}^{W}-w\left(i,j\right)\left[y_{i,j}\log p_{i,j}+\left(1-y_{i,j})\log\left(1-p_{i,j}\right)\right)\right]\;,$$
where y is the ground truth, p is the predict logits, *i* and *j* are the pixel index in the image.

## 4. Experiments

In this section, we provide qualitative and quantitative results for experiments carried out to test the performance of our approach. There are numerous approaches in semantic segmentation; we mainly compare our method to those pursuing a good tradeoff between high quality and computation, such as SegNet, ENet, and ERFNet. Moreover, to compare [22], we verified the model of inferring occluded road boundaries by replacing the decoder part of the model with a new one that is suitable for our task. The verified model is referred to as ORBNet in our work, which retained the encoder and employed a decoder similar to that in the DeepLabv3+ algorithm [6]. We present quantitative results based on evaluations with our manually annotated dataset based on the KITTI dataset named KITTI-OFRS dataset. The presented results appear all to be based on the manual dataset annotations except the qualitative results on Cityscapes dataset using predicted semantics as input. We first trained the models on the proposed KITTI-OFRS dataset, and the experimental results demonstrate that the proposed approach spends less time on inference and obtains better performance. Then, we compared the performance of those models when trained with traditional cross-entropy loss function and the proposed spatially-weighted cross-entropy loss function. Moreover, we tested the generalization performance of the models on the Cityscapes dataset. Finally, the performance of the models based on automatically inferred semantics was visualized to show that our network works well in the real system.

### 4.1. Datasets

There were no available datasets for the proposed occlusion-free road segmentation task, so we built our own datasets. We built a real-world dataset named KITTI-OFRS based on the public KITTI semantic segmentation benchmark, which is used for training and evaluation. Moreover, we qualitatively tested our well-trained model on the Cityscape dataset [33] for a view of its generalization ability.

**KITTI-OFRS Dataset** The real-world dataset was built on the public KITTI semantic segmentation benchmark, which is part of the KITTI dataset [34]. The KITTI dataset is the largest data collection for computer vision algorithms in the world’s largest autopilot scenario. The dataset is used to evaluate the performance of computer vision technologies and contains real-world image data collected from scenes such as urban, rural, and highways, with up to 15 vehicles and 30 pedestrians per image, as well as varying degrees of occlusion. The KITTI semantic segmentation benchmark consists of 200 semantically annotated train as well as 200 test samples corresponding to the KITTI Stereo and Flow Benchmark 2015. We only annotated the available 200 semantically annotated training samples for our task and randomly split them into two parts, one contained 160 samples for training, and the other contained 40 samples for evaluation. We named this real-world dataset as KITTI-OFRS dataset. One sample in this dataset contained the RGB image, normal semantic labels, and occlusion-free road segmentation labels, as demonstrated in Figure 7.

**Cityscapes Dataset** The Cityscapes dataset contains 5000 images collected in street scenes from 50 different cities. The dataset is divided into three subsets, including 2975 images in the training set, 500 images in the validation set, and 1525 images in the testing set. High-quality pixel-level annotations of 19 semantic classes are provided in this dataset. We only used this dataset for the generalization ability test.

**Classes Transformation** The occlusion-free road segmentation network was designed to apply in the semantic domain. However, different semantic segmentation datasets may have different categories, and one category may have a different class labels in different datasets. It is obvious that some categories are not involved in occluding the road, such as sky, and some categories could be aggregated to one category to get a more compact representation, for example, car, truck, bus, train, motorcycle, and bicycle could be aggregated to vehicle. Therefore, a classes transformation layer is proposed to transform different semantic representations to a unify form before being fed to the occlusion-free road segmentation network. 

The classes transformation layer is a matrix multiplication operation, taking one-hot liked encoded semantic representation of variable categories $R_{in}\in {\left[0,1\right]}^{W\times H\times C}$ as input and output one-hot representation of a unify categories $R_{out}\in {\left[0,1\right]}^{W\times H\times C_{u}}$.
(5)$$R_{out}=R_{in}*T,$$
(6)$$T\left(i,j\right)=\{\begin{array}{cc} 1, & if\;C\left(i\right)\to C_{u}\left(j\right)\\ 0, & otherwise \end{array},$$
where $T\in {\left\{0,1\right\}}^{C\times C_{u}}$ is the transformation matrix, C is the set of original class labels and $C_{u}$ the set of target class labels. $C\left(i\right)\to C_{u}\left(j\right)$ refers to that the *i*-th label in C should be set to the *j*-th label in $C_{u}$.

The classes transformation layer could aggregate and unify labels of different semantic segmentation representations from different datasets or different semantic segmentation algorithms. In our work, the unified semantic representation contained 11 classes, namely road, sidewalk, building, wall, fence, pole, traffic sign, vegetation, person, vehicle, and unlabeled. 

**Data Augmentation** In the training phase, the training data was augmented with random cropping and padding, flipping left to right. Moreover, to tackle the uncertainty of the semantic labels due to annotation errors, we augmented the training data by the technique of label smoothing, which is firstly proposed in InceptionV2 [35] to reduce over-fitting and increase the adaptive ability of the model. We used this method to add noise to the semantic one-hot, which could make our model more adaptive to annotation errors and prediction errors from other semantic segmentation methods. Unlike the original usage that takes α a constant value for all the samples, we choose α as a random value between 0.1 and 0.2 following uniform distribution, which was independent of each pixel in a training batch.

(7)$$y_{k}^{\mathrm{LS}}=y_{k}+\left(1-\alpha \right)+\alpha /K.$$

### 4.2. Evaluation Metrics

For quantitative evaluation, precision (PRE), recall (REC), F1 score, average precision (ACC), and intersection-over-union (IoU) were used as the metrics within a region around the road edges within 4 pixels. The metrics acting on such a region are more powerful to test the network performance than on the whole pixels taking into account the primary task of occlusion reasoning. The metrics are calculated as in Equations (8)–(12), where TP, TN, FP, FN are, respectively, the number of true positives, true negatives, false positives, and false negatives at the pixel level. Our experiments considered an assessment that demonstrates the effectiveness of our approach for inferring occluded road in the semantic domain. 

(8)$$\mathrm{PRE}=\frac{TP}{TP+FP}\;,$$

(9)$$\mathrm{REC}=\frac{TP}{TP+FN}\;,$$

(10)$$F1=\frac{2PRE\cdot REC}{PRE+REC}\;,$$

(11)$$\mathrm{ACC}=\frac{TP+TN}{TP+FP+TN+FN}\;,$$

(12)$$\mathrm{IoU}=\frac{TP}{TP+FP+FN}\;.$$

### 4.3. Implementation Details

In the experiments, we implemented our architectures in PyTorch version 1.2 [36] (FaceBook, State of California, USA) with CUDA 10.0 and cuDNN back-ends. All experiments were run on a single NVIDIA GTX-1080Ti GPU. Due to GPU memory limitations, we had a maximum batch size of 4. During optimization, we used the SGD optimizer [37] with a weight decay of 0.0001 and a momentum of 0.9. The learning rate was set using the poly strategy with a start value of 0.01 and a power of 0.9. The edge region width $T_{w}$ was set to 10 in the training phase and 4 in the evaluation phase.

### 4.4. Results and Analysis

To evaluate the effectiveness of our method on the occlusion-free road segmentation task, we trained the proposed model on the KITTI-OFRS dataset, as well as some other lightweight baseline models, such as ENet, SegNet, ERFNet, and ORBNet. The samples were resized to 384 × 1248 when training and testing. The quantitative and qualitative results are shown in Table 2 and Figure 8, respectively. As shown in Table 2 and Figure 8, both models achieved comparable results on the proposed task, and our method was superior to the baseline models in both accuracy and runtime. In Figure 8, red denotes false negatives; blue areas correspond to false positives, and green represents true positives. The models both performed well in the semantic domain containing more compact information of the driving environment, which indicates that the semantic and spatial information were more essential for occlusion reasoning than color and textural features. As can be seen from Figure 8, the models obtained significant results on both simple straight roads and complex intersection areas. Variable occlusion situations could be handled well, even though there were some heavy occlusion scenes. Based on the results of the proposed task, the whole road structure could be obtained and could be easily transformed into 3D world representations by an inverse perspective transformation without the affectation of the on-road objects. Empirically, higher road detection precision may lead to a better road model for better path planning.

**Comparison of accuracy and computation complexity** Our model achieved a significant trade-off between accuracy and efficiency, which conclusion is drawn by comparing with other models. To compare the computation complexity, we employed several parameters, GFLOPs, and frames per second (FPS) as the evaluation metrics. FPS was measured on an Nvidia GTX1080Ti GPU with an input size of 384 × 1248 and was averaged among 100 runs. As can be seen from Table 2, our model outperformed ENet by 1.5% in the F1 score and 2.6% in the IoU while runs were only a little slower than it. Our model ran almost two times faster than ERFNet and improved 1.0% in the F1 score and 1.7% in the IoU. Compared to SegNet and ORBNet, our model got a little improvement in accuracy but achieved three times faster in the inference phase. In conclusion, our model achieved a better trade-off between accuracy and efficiency. 

**Comparison of loss function** To evaluate the effectiveness of the proposed spatially-weighted cross-entropy loss, we trained the models both with traditional cross-entropy loss (CE) and the spatially-weighted cross-entropy loss (CE-SW), and the evaluation results of the CE and metrics degradation are shown in Table 3. When trained with CE, the models saw obvious metrics degradation compared to CE-SW. The values in parentheses are the metrics degradation compared to that when models were trained with CE-SW, which shows that the spatially-weighted cross-entropy loss was very beneficial for increasing accuracy. Intuitively, the spatially-weighted cross-entropy loss forced the models to take care of the road edge region where the occlusion occurs mostly.

**Comparison of convolution with and without context** To evaluate the benefits of the context convolution block, we replaced the context convolution block with regular convolution operation in the down-sampling and up-sampling blocks. As shown in Table 4, the model with context information outperformed the model without that by 0.6% in the F1 score and 1.0% in the IoU, which demonstrates that the context information is desirable for the proposed approach.

**Generalization on Cityscape Dataset** To further test the generalization ability of our model, we conducted qualitative test experiments on the Cityscape dataset with the model trained only on the KITTI-OFRS dataset. As can be seen from Figure 9, the well-trained model performed well on the complex real-world Cityscapes dataset, which indicates that our model obtained quite a good generalization ability on the occlusion-free road segmentation task. The generalization ability of our model benefited from inferring the occluded road in the semantic domain, which made the model focus on learning the occlusion mechanism in the driving scenes without the affectation of sensing noise. In the scenes, the color and textual features may differ very much in the same position due to different camera configurations and lighting conditions while the semantic features share a similar distribution. The occlusion situations were able to understand that the occluded road area was correctly inferred in variable occlusion scenes by the proposed method according to the results. As shown in Figure 9, the detection results obtained the overall structure of the road and accurate segmentation despite occlusion. Moreover, it is applicable to combine our method with other semantic segmentation algorithms in the real system due to its lightweight and efficiency. As shown in Figure 10, when taking the predicted semantics obtained by the DeepLabv3+ algorithm as input, the proposed OFRSNet still works well to predict the occluded road areas and outperforms ENet and ORBNet in terms of accuracy and robustness. 

## 5. Conclusions

In this paper, we presented an occlusion-free road segmentation network to infer the occluded road area of an urban driving scenario from monocular vision. The model we presented is a lightweight and efficient encoder–decoder fully convolutional architecture that contains down-sampling and up-sampling blocks combined with global contextual operations. Meanwhile, a spatially-weighted cross-entropy loss was proposed to induce the network to pay more attention to the road edge region in the training phase. We showed the effectiveness of the model on the self-built small but efficient KITTI-OFRS dataset. Compared to other recent lightweight semantic segmentation algorithms, our network obtained a better trade-off between accuracy and runtime. The comparisons of the models trained with different loss functions highlighted the benefits of the proposed spatially-weighted cross-entropy loss for the occlusion reasoning road segmentation task. The generalization ability of our model was further qualitatively tested on the Cityscape datasets, and the results clearly demonstrated our model’s inferring ability of the occluded road even in complex scenes. Moreover, the proposed OFRSNet could be efficiently combined with other semantic segmentation algorithms due to its small size and minimal runtime. We believe that being able to infer occluded road regions in autonomous driving systems is a key component to achieve a full comprehension of the scene and will allow better planning of the ego-vehicle trajectories.

## Figures and Tables

**Figure 1 sensors-19-04711-f001:**
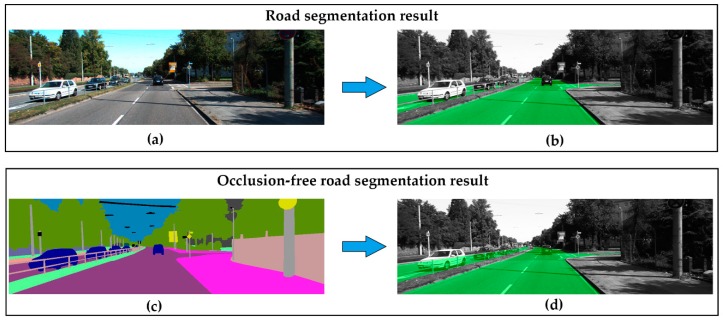
Comparison of road segmentation and proposed occlusion-free road segmentation. (**a**) RGB image; (**b**) visualization of the results of road segmentation; (**c**) visualization of the semantic representation of the scene, which could be obtained by semantic segmentation algorithms in real applications or human annotation in training phase; (**d**) visualization of the results of the proposed occlusion-free road segmentation. Green refers to the road area in (**b**) and (**d**).

**Figure 2 sensors-19-04711-f002:**
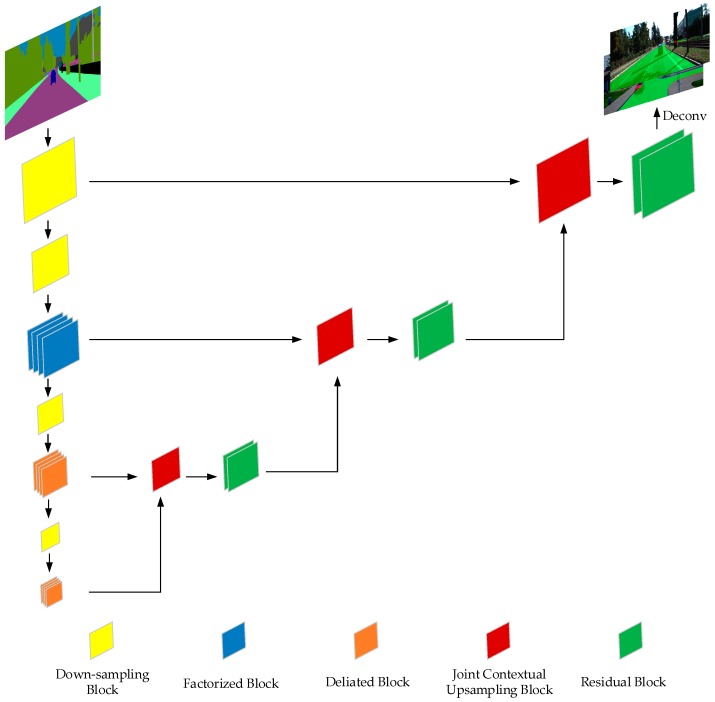
The proposed occlusion-free road segmentation network architecture.

**Figure 3 sensors-19-04711-f003:**
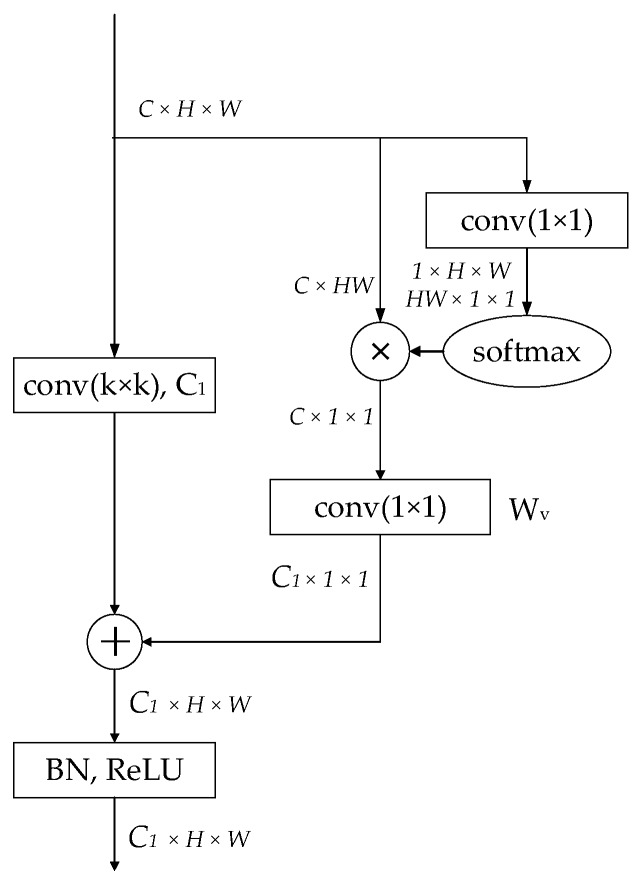
The context convolution block.

**Figure 4 sensors-19-04711-f004:**
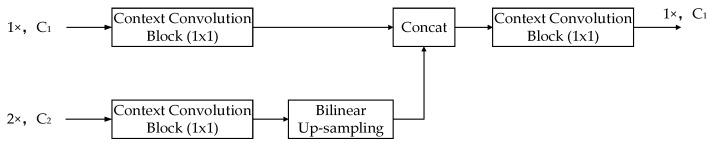
The joint context up-sampling block.

**Figure 5 sensors-19-04711-f005:**
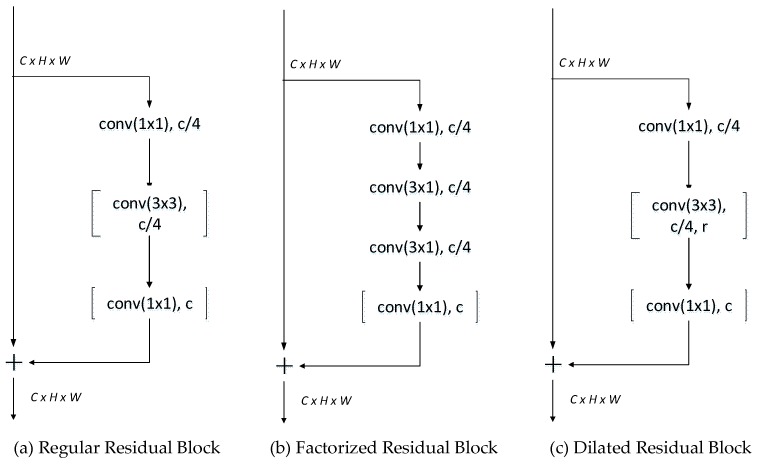
Residual blocks in our network.

**Figure 6 sensors-19-04711-f006:**
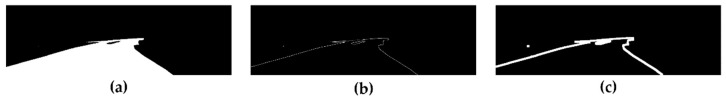
Visualization of the road edge region. (**a**) The road segmentation label; (**b**) road edge obtained from (a) by the Canny algorithm; (**c**) road edge region with a width of 10 pixels.

**Figure 7 sensors-19-04711-f007:**
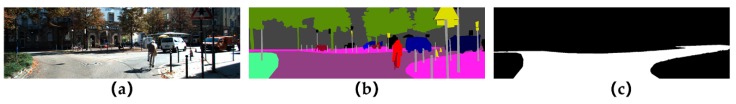
An example of the KITTI-occlusion-free road segmentation (KITTI-OFRS) dataset sample. (**a**) the RGB image; (**b**) annotation of semantic segmentation; (**c**) annotation of full road area, white denotes road.

**Figure 8 sensors-19-04711-f008:**
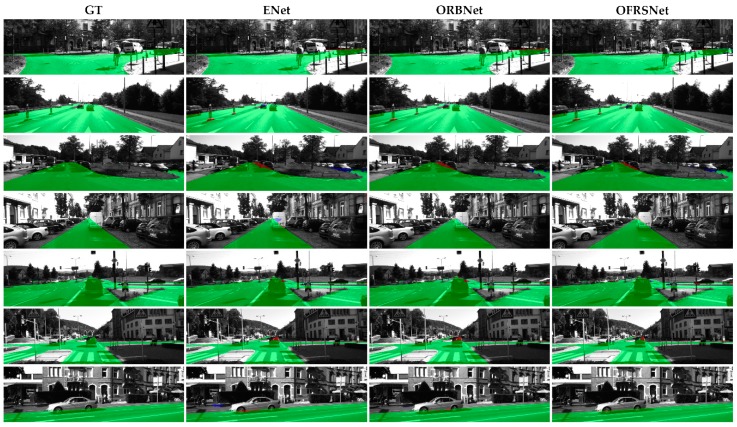
Qualitative results on the KITTI-OFRS dataset. The columns from left to right are the results of GT, ENet, ORBNet, and OFRSNet, respectively. Red denotes false negatives; blue areas correspond to false positives, and green represents true positives.

**Figure 9 sensors-19-04711-f009:**
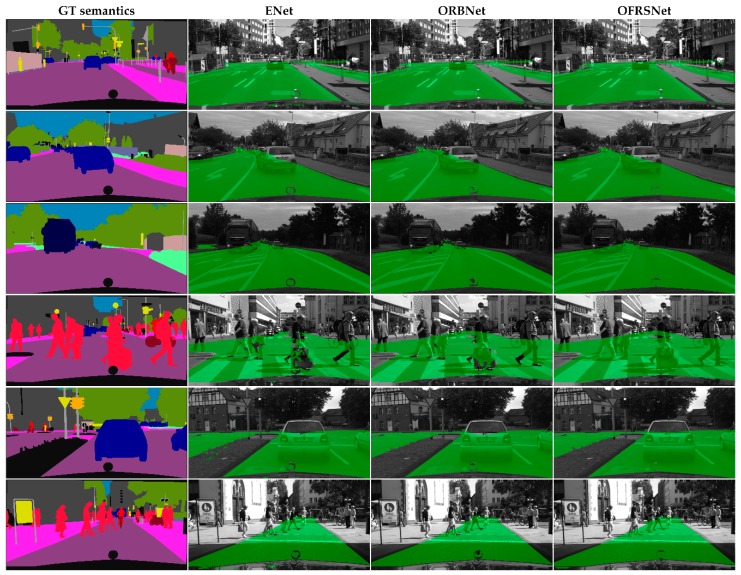
Qualitative results on the Cityscapes dataset using ground truth semantics as input. Green represents the detected full road area.

**Figure 10 sensors-19-04711-f010:**
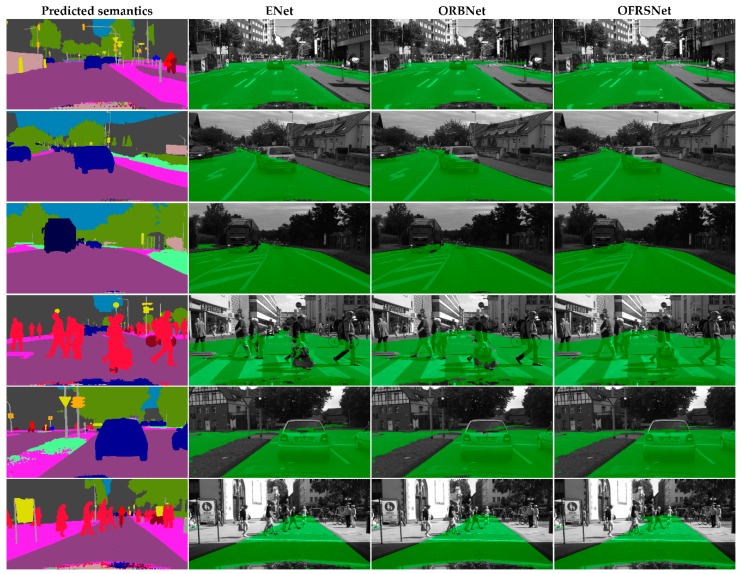
Qualitative results on the Cityscapes dataset using predicted semantics as input, which were obtained by the DeepLabv3+ algorithm. Green represents the detected full road area.

**Table 1 sensors-19-04711-t001:** Our network architecture in detail. Size refers to output feature maps size for an input size of 384 × 1248.

Stage	Block Type	Size
Encoder	Context Down-sampling	192 × 624 × 16
	Context Down-sampling	96 × 312 × 32
	Factorized blocks	96 × 312 × 32
	Context Down-sampling	48 × 156 × 64
	Dilated blocks	48 × 156 × 64
	Context down-sampling	24 × 78 × 128
	Dilated blocks	24 × 78 × 128
Decoder	Joint Context Up-sampling	48 × 156 × 64
	Bottleneck Blocks	48 × 156 × 64
	Joint Context Up-sampling	96 × 312 × 32
	Bottleneck Blocks	96 × 312 × 32
	Joint Context Up-sampling	192 × 624 × 16
	Bottleneck Blocks	192 × 624 × 16
	Deconv	384 × 1248 × 2

**Table 2 sensors-19-04711-t002:** Evaluation results of models trained with spatially-weighted cross-entropy loss (CE-SW).

Model	Parameters	GFLOPs	FPS	ACC	PRE	REC	F1	IoU
ENet	0.37M	3.83	**52**	91.8%	92.1%	89.3%	90.7%	82.9%
ERFNet	2.06M	24.43	25	92.3%	92.6%	89.7%	91.2%	83.8%
SegNet	29.46M	286.03	16	92.9%	93.6%	90.2%	91.8%	84.9%
ORBNet	1.91M	48.48	11.5	92.7%	93.4%	89.9%	91.6%	84.5%
OFRSNet	0.39M	2.99	46	**93.2%**	**94.2%**	**90.3%**	**92.2%**	**85.5%**

**Table 3 sensors-19-04711-t003:** Evaluation results of models trained with cross-entropy loss (CE). The values in parentheses are the metrics degradation compared to that when models were trained with spatially-weighted cross-entropy loss (CE-SW).

Model	ACC	PRE	REC	F1	IoU
ENet	90.4%(−1.4%)	90.5%(−1.6%)	87.6%(−1.7%)	89.0%(−1.7%)	80.2%(−2.7%)
ERFNet	90.5%(−1.8%)	90.9%(−1.7%)	87.3%(−2.4%)	89.1%(−2.1%)	80.3%(−3.5%)
SegNet	92.1%(−0.8%)	92.6%(−1.0%)	89.4%(−0.8%)	91.0%(−0.8%)	83.5%(−1.4%)
ORBNet	91.5% (−1.2%)	92.2% (−1.2%)	88.4% (−1.5%)	90.2% (−1.4%)	82.2% (−2.3%)
OFRSNet	91.7%(−1.5%)	92.4%(−1.8%)	88.6%(−1.7%)	90.5%(−1.7%)	82.6%(−2.9%)

**Table 4 sensors-19-04711-t004:** Performance comparison of the model with and without context.

Model	Context	Parameters	GFLOPs	ACC	PRE	REC	F1	IoU
OFRSNet	w/o	0.34M	2.96	92.7%	92.8%	90.4%	91.6%	84.5%
OFRSNet	w/	0.39M	2.99	**93.2** **%**	**94.2** **%**	**90.3** **%**	**92.2** **%**	**85.5** **%**

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
