# Peer review of "Occlusion-Free Road Segmentation Leveraging Semantics for Autonomous Vehicles"

_sensors, 2019, doi:10.3390/s19214711_

Round 1

Reviewer 1 Report

Paper describes an approach to road segmentation that takes a multi-label semantic image as input and predicts binary occlusion-free road segmentation. Based on this I suggest the title should be "Occlusion-Free Road Segmentation...." (OFRS) for clarity.

Authors provide a good comparison with similar deep net architectures (tab.2), which shows some marginal improvement (+2% edge accuracy) by trading-off speed (-10% fps) compared to the most similar model (ENet). This tweak is facilitated by up-weighting the loss near detected semantic edges and horizon areas. In practice, the gain is however barely noticeable (fig.8) and I doubt it has any impact on the application like path planning (which should be discussed in the paper).

Overall, with accuracy over 90%, the road segmentation problem seems largely solved by the state of the art already, and the conclusion is over-selling the results in this aspect.

More interesting would be some integration with the base multi-class semantic segmentation from RGB images. I however welcome the idea of having a dedicated dataset based on the popular KITTY, which turns the submission somehow into a benchmark paper.

To make this complete, further comparison with the specific state of the art [22,23] on the new dataset, or at least the qualitative case on Cityscapes.

The presented results appear all to be base on the manual dataset annotations, but this should be clarified in the Experiments section. Also, results based on automatically inferred semantics should be included to give some idea how this all would work on a real system.

In summary, further above mentioned experimental comparison must be included before considering publication of this paper; this is essential for all empirical deep-learned methods.

eq.(2)/(4) use of the same symbol "p" with a different meaning
l.208 ->Canny
l.271 unique->uniform?
l. 310 "runs as fast as ENet" - not true according to tab.2 (46 vs. 52 fps)
fig. 8 use greyscale (not colour) images under the colour-coded segmentation
fig. 8 add GT column
fig. 9 add basic ENet and [23]

Reviewer 2 Report

The authors present a method for occlusion reasoned segmentation for autonomous vehicles.

Their introduction and bibliography search is satisfactory but they can improve their related work section.   Their methodology is explained clearly and well, while the experiments are also well designed. Their results and validation are supporting their conclusions.

The manuscript requires moderate English changes.

For example in line 51 the should add “by the fact” after the word inspired.

Line 53 network could learns need to change to network could learn

The caption in figure 3 (c) should write white instead of write

Reviewer 3 Report

This paper proposes a fully convolutional neural network based approach to infer the occluded road area for urban driving scenarios form monocular vision.

Overall, this paper is well-written and the proposed approach is promising. Thus, publication is recommended.

In particular, it clearly presented the architecture of the proposed CNN with enough justifications for the design selection. It also uses a new loss function with spatial-dependent weight to enhance detection accuracy. Finally, detailed experimental results and analysis are presented to demonstrate the advantages of the proposed algorithm.

Minor typos might exist in the paper. For example, in Figure 7 title, “write” should be “white”.

Round 2

Reviewer 1 Report

Authors improved the text by improving experiments and additional comparison, which shows the proposed method has advantage on real data. Based on this I consider the paper suitable for publication in Sensors, after minor correction below.

There was some misunderstanding about my suggestion for the use of colours in Fig.8, which should be composed of two blended layers as follows:

1) Background = greyscale intensity image (simply convert original RGB image to single-channel image).

2) Overlay = colour semantic (green/blue/red) - revert to how it was in v1, including FP/FN

Only in this way you can safely distinguish where the colour comes from (eg. road semantics vs. green part of a tree). 

Author Response

This manuscript is a resubmission of an earlier submission. The following is a list of the peer review reports and author responses from that submission.